# Methods for the Inclusion of Real-World Evidence in a Rare Events Meta-Analysis of Randomized Controlled Trials

**DOI:** 10.3390/jcm12041690

**Published:** 2023-02-20

**Authors:** Minghong Yao, Yuning Wang, Fan Mei, Kang Zou, Ling Li, Xin Sun

**Affiliations:** 1Chinese Evidence-Based Medicine Center and MAGIC China Center, West China Hospital, Sichuan University, Chengdu 610041, China; 2NMPA Key Laboratory for Real World Data Research and Evaluation in Hainan, Chengdu 610041, China; 3Sichuan Center of Technology Innovation for Real World Data, Chengdu 610041, China

**Keywords:** meta-analysis, rare events, real-world evidence, randomized controlled trial

## Abstract

Background: Many rare events meta-analyses of randomized controlled trials (RCTs) have lower statistical power, and real-world evidence (RWE) is becoming widely recognized as a valuable source of evidence. The purpose of this study is to investigate methods for including RWE in a rare events meta-analysis of RCTs and the impact on the level of uncertainty around the estimates. Methods: Four methods for the inclusion of RWE in evidence synthesis were investigated by applying them to two previously published rare events meta-analyses: the naïve data synthesis (NDS), the design-adjusted synthesis (DAS), the use of RWE as prior information (RPI), and the three-level hierarchical models (THMs). We gauged the effect of the inclusion of RWE by varying the degree of confidence placed in RWE. Results: This study showed that the inclusion of RWE in a rare events meta-analysis of RCTs could increase the precision of the estimates, but this depended on the method of inclusion and the level of confidence placed in RWE. NDS cannot consider the bias of RWE, and its results may be misleading. DAS resulted in stable estimates for the two examples, regardless of whether we placed high- or low-level confidence in RWE. The results of the RPI approach were sensitive to the confidence level placed in RWE. The THM was effective in allowing for accommodating differences between study types, while it had a conservative result compared with other methods. Conclusion: The inclusion of RWE in a rare events meta-analysis of RCTs could increase the level of certainty of the estimates and enhance the decision-making process. DAS might be appropriate for inclusion of RWE in a rare event meta-analysis of RCTs, but further evaluation in different scenarios of empirical or simulation studies is still warranted.

## 1. Introduction

Well-conducted randomized controlled trials (RCTs) have been considered the gold standard, and as such are often used in meta-analyses to evaluate the effects of healthcare interventions [1]. For rare events meta-analyses of RCTs, the outcomes of the included trials may be very sparse, with no events even being observed in some trials [2,3], resulting in lower statistical power [4]. Recently, there has been a growing interest in the use of real-world evidence (RWE) in clinical drug assessments and health care evaluations [5,6,7]. RWE is clinical evidence from multiple sources outside the typical clinical research setting, including electronic health records and billing databases [5]. Regulatory bodies such as the US Food and Drug Administration (FDA) have been increasingly using RWE to monitor adverse post-market safety events and make regulatory decisions [5,8]. An important potential advantage of RWE is that it can represent real-life clinical practice across a broader and more diverse distribution of patients than RCTs [9]. The results from RWE studies, such as those obtained from epidemiology databases, health surveys, and electronic medical records, typically include larger sample sizes and have longer follow-up periods, increasing the probability of finding rare events [10,11,12].

RCTs and RWE studies are potentially valuable evidence sources for assessing the effect of rare events, and integrating RWE studies and RCTs to assess the effects of rare events may help increase the overall level of certainty of evidence [13,14]. However, the inclusion of RWE in a rare events meta-analysis of RCTs is not a straightforward issue, as the estimates obtained from RWE studies may be subject to bias due to the potential selection and information biases of real-world data [15]. Therefore, RCTs and RWE studies should not be directly integrated without considering the bias of the RWE studies. Generalized evidence synthesis statistical approaches for integrating RCTs and RWE studies can provide not only an integrated result but also a quantitative analysis of the influence of RWE on the integrated evidence under different bias assumptions for the RWE studies, which is more attractive for decision-making [16,17,18,19]. Relevant methods have been published in the statistical literature but have not been systematically verified or widely used in empirical analyses [16].

Therefore, the aim of this study is to investigate whether these statistical methods, including naïve data synthesis (NDS), design-adjusted synthesis (DAS), the use of RWE as prior information (RPI), and three-level hierarchical models (THMs), could be used for integrating RWE in a rare events meta-analysis of RCTs by considering the potential inherent biases of RWE studies and their impact on the level of uncertainty around the estimates.

## 2. Materials and Methods

### 2.1. Study Design Overview

In this study, we used four approaches to combine RCTs and RWE studies [18,20]: (1) NDS, where data from all studies, regardless of the study design, were directly combined; (2) DAS, where RCTs and RWE studies were synthesized and where the information from the RWE studies was adjusted to reflect the confidence in the study findings; (3) the RPI, which included RWE via the prior for the treatment effect and combined this with a likelihood based only on the data from RCTs; (4) the THM, which was used to simultaneously model the between-study heterogeneity of treatment effects within each study design (RCT or RWE studies) and across study designs. The methodologies were applied to two recently published meta-analyses concerning the risk of diabetic ketoacidosis (DKA) among patients using sodium/glucose cotransporter 2 (SGLT-2) inhibitors compared with active comparators [21] and the effectiveness of 23-valent pneumococcal polysaccharide vaccine (PPV23) vaccination against invasive pneumococcal disease (IPD) in elderly patients [22]. Here, we describe how these methods can be performed and how the bias of the RWE studies can be considered. Finally, we discuss the pros and cons of each method.

### 2.2. Illustrative Example Dataset

The first example was a recent meta-analysis conducted by Falkenhorst et al. [22]. In this meta-analysis, the authors used the evidence from RCTs and RWE studies to investigate the effectiveness of PPV23 vaccination against IPD and pneumococcal pneumonia in adults aged ≥ 60 years. Here, we focus on IPD. This study included four RCTs, five cohort studies, three case-control studies, and five case–case studies. We excluded one RCT (for reporting no events in both treatment groups) and five case–case studies (for IPD not being reported) in our study. The outcomes of these studies are displayed in Table 1. The authors assessed the risk of bias of individual studies using the Cochrane Risk of Bias tool [23] for RCTs and the Newcastle-Ottawa Scale for observational studies [24]. A total of 16 IPD events were reported across three RCTs, which randomized a total of 41,992 patients, and the incidence rate was 0.04%. We observed 165 IPD events among 575,454 patients in five cohort studies.

The second example showed how to integrate RCTs and RWE studies when assessing the effects of rare adverse events. A typical example of rare adverse events is the risk of DKA caused by SGLT-2 inhibitors. The data used in this study were obtained from the study conducted by Alkabbani et al. [21]. This study included twelve placebo-controlled RCTs, seven active comparator RCTs, and seven active comparator RWE studies. All were retrospective propensity score-matched cohort studies. Our primary concern was whether SGLT-2 inhibitors increase the risk of DKA compared with active comparators. We excluded one RWE study because its control was not an active comparator [36]. In this study, the authors assessed the risk of bias in each study using the checklist proposed by Downs et al. [37] for RCTs and RWE studies. Table 2 shows the basic characteristics of the included RCTs and RWE studies. A total of 8 DKA events were reported across all RCTs, which randomized 8100 patients, and the incidence rate was 0.1%. We observed 2016 DKA events among 1,072,992 patients across all RWE studies.

### 2.3. Methods for Incorporating RWE in a Rare Events Meta-Analysis of RCTs

In the following, we describe how these methods can be performed and how the bias of the RWE studies can be considered. We used *j* = 1, 2, … k to denote study *j*. The odds ratio (OR) metric was used as the effect measure because rare events are often binary outcomes. These methods are also suitable for other measures, such as the relative effect (RR) and risk difference (RD).

#### 2.3.1. The Naïve Data Synthesis

The NDS is identical to the conventional random effects meta-analysis model, namely the normal–normal hierarchical model. The model was then written as follows:$$d_{j}\sim N\left(\theta_{j},s_{j}^{2}\right)$$
$$\theta_{j}\sim N\left(\theta ,\tau^{2}\right)$$
where $d_{j}$ and $s_{j}^{2}$ are the study-specific empirical effect and the within-study variances for study *j*, respectively. Both the treatment effect ($d_{i}$) and variance ($s_{i}^{2}$) are on the log OR scale. Here, $\theta_{j}$ denotes the true underlying log-OR for the *j*th study, θ is the summary log-OR, and $\tau^{2}$ is the between-study variance. We assigned a weakly informative prior for θ, i.e., $\theta \sim N\left(0,2.82^{2}\right)$, following Günhan et al. [49]. For the heterogeneity parameter τ, we used a half-normal prior with a scale of 0.5 (τ~HN(0.5)), following Friede et al. [50].

#### 2.3.2. The Design-Adjusted Synthesis

The DAS approach was originally used for integrating RCTs and RWE studies in network meta-analysis. We simplified this model to fit a pairwise meta-analysis. This model gives less weight to RWE studies compared to RCTs. This down-weighting is achieved by inflating the variance of the mean effect of RWE studies. The general conventional meta-analysis model was extended by including a down weighting factor $\omega_{j}$ in the variance of the estimated treatment effect of study *j*:$$d_{j}\sim N\left(\theta_{j},\frac{s_{j}^{2}}{\omega_{j}}\right)$$
$$\theta_{j}\sim N\left(\theta ,\tau^{2}\right)$$
where $\omega_{j}$ is the variance inflation factor for study *j,* and the other parameters are the same as the DAS; $\omega_{j}$ was equal to 1 when the study type was an RCT, since the estimates obtained from RCTs are often considered unbiased. For the RWE studies, the $\omega_{j}$ values ranged from zero to one, and the variance of the treatment effect was inflated so that the weight in the meta-analysis was decreased. We evaluated varying levels of trust in the RWE studies, with $\omega_{j}$~beta (4, 1), $\omega_{j}$~beta (1.5, 1), and $\omega_{j}$~beta (0.25, 1) standing for high, medium, and lower levels of confidence in RWE studies. The prior specifications for θ and τ were identical to those of the NDS.

#### 2.3.3. The Use of Real-World Evidence as Prior Information

This approach included RWE via the prior for the treatment effect and combined this with a likelihood based only on the data from RCTs. We first synthesized the RWE studies to generate ${\hat{\theta }}_{RWE}$ with a corresponding variance ${\hat{V}}_{RWE}$. Then, we centered the informative prior for the population mean (θ) on ${\hat{\theta }}_{RWE}$ but use an inflated variance of the mean effect of RWE; that is, $\theta \sim N\left({\hat{\theta }}_{RWE},\frac{{\hat{V}}_{RWE}}{\omega }\right)$, where ω is the variance inflation factor. The prior specification for τ is identical to the NDS. We also evaluated varying levels of trust in the RWE studies, with $\omega_{j}$~beta (4, 1), $\omega_{j}$~beta (1.5, 1), and $\omega_{j}$~beta (0.25, 1).

#### 2.3.4. The Three-Level Hierarchical Model

For the THM, the evidence from the RCTs and RWE studies were modeled separately at the within-study and within-design levels, and then the estimates for RCTs and RWE studies were combined in an overall measure of the treatment effect using the random-effects model, assuming *i* = 1, 2, where 1 represents the RCTs and 2 represents the RWE studies. The model is presented as follows:$$d_{j,i}\sim N\left(\theta_{j,i},s_{j,i}^{2}\right)$$
$$\theta_{j,i}\sim N\left(\theta_{i},\tau_{i}^{2}\right)$$
$$\theta_{i}\sim N\left(\theta ,\tau^{2}\right)$$
where $d_{j,i}$ and $s_{j}^{2}$ are the study-specific empirical effect and the within-study variances for study j and design *i*, respectively; $\theta_{j,i}$ denotes the true underlying effect for the *j*th study and *i*th design; $\theta_{i}$ and $\tau_{i}^{2}$ are the summary effect and the variance for the *i*th design, respectively; and the other parameters are the same as the NDS. The RWE studies could be down-weighted by inflating the variance; that is, θ2~N(θ,τ2ω). The prior specifications for θ and τ were identical to those of the NDS. For the study-design variance $\tau_{i}^{2}$, we also assigned a half-normal prior with a scale of 0.5. The prior specifications for ω were also set in three different scenarios, namely $\omega_{j}$~beta (4, 1), $\omega_{j}$~beta (1.5, 1), and $\omega_{j}$~beta (0.25, 1).

### 2.4. Implementation and Model Fit

For RCTs, the effect of the single-zero event trial was estimated by the method proposed by Greenland et al. [51]. All models were performed using Hamiltonian Monte Carlo algorithms, as implemented in the *RStan* package (version 2.21.2) in the R statistical environment (version 4.0.3, R Foundation for Statistical Computing, Vienna, Austria). As a comparison, we estimated the results of RCTs using a Bayesian random effects meta-analysis. We fitted four chains for each model, each with 5000 iterations, the first half of which were considered warmups and discarded. Convergence was judged to have occurred when $\hat{R}$ (the potential scale reduction factor) was no greater than 1.1 for all parameters [52].

## 3. Results

### 3.1. Effectiveness of PPV23 Vaccination against IPD in Elderly Patients

The Bayesian meta-analysis including only RCTs had a posterior median of 0.29 and a 95% credibility interval (CrI) [0.06, 1.45], which indicated that based on the RCT evidence, the PPV23 is not effective against IPD. The posterior median and 95% CrI based on the NDS were 0.42 and (0.29, 0.58), which indicated the effectiveness of PPV23 against IPD in elderly patients. Table 3 shows the posterior median and 95% CrI of the effectiveness of PPV23 vaccination against IPD in elderly patients using the DAS, RPI, and THM. The models, except for the THM, consistently showed the effectiveness of PPV23 vaccination against IPD in elderly patients. By comparing the length of the intervals between the NDS and the other three models, we found that the length of the interval of the NDS was much smaller than that of the other three models. We also observed that the length of the interval in the estimates obtained from the four models was much smaller than the estimates from only RCTs.

### 3.2. Risk of DKA among Users Receiving SGLT-2 Inhibitors Versus Active Comparators

We did not observe an increased risk of DKA when the results were pooled from the RCTs (OR = 1.02, 95% CrI: 0.37–2.86), while we observed an increased risk of DKA when directly combining the RCTs and RWE studies using NDS (OR = 1.58, 95% CrI: 1.22–2.04). The estimates of each model under different variance inflation factors, which represent different levels of confidence in RWE studies, are presented in Table 4. The results of the different models were inconsistent. The DAS showed that SGLT2 inhibitors were associated with an increased risk of DKA under different confidence levels placed in RWE studies. For the RPI, we observed an increased risk when we placed a high level of confidence in RWE studies (OR = 1.56, 95% CrI: 1.12–2.12); however, we did not observe an increased risk when we placed a medium level of confidence (OR = 1.52, 95% CrI: 0.98–2.19) or a low level (OR = 1.41, 95% CrI: 0.56–2.45) of confidence in RWE studies. We also did not observe an increased risk of DKA by using the THM, even though we placed a high level of confidence in RWE studies (OR = 1.42, 95% CrI: 0.50–3.75). As with the first example, the interval lengths of the four model estimates were smaller than the outcomes estimated based on RCTs alone.

## 4. Discussion

In this study, we discussed approaches for incorporating RWE in a rare events meta-analysis of RCTs, with particular interest in the effectiveness of PPV23 against IPD in elderly patients and the risk of DKA among patients using SGLT-2 inhibitors compared with active comparators. We did not observe the effectiveness of PPV23 and the risk of DKA among patients using SGLT-2 inhibitors when the data were only from RCTs. In the effectiveness example of PPV 23, all methods, except for the THM, found that the inclusion of RWE studies in rare events meta-analysis of RCTs could complement the evidence of the findings of the RCT-only analysis. However, for the risk of DKA among patients using SGLT-2 inhibitors, the results of the different models were inconsistent. For instance, for the RPI, an obvious finding was that we did not observe an increased risk when low and medium confidence was placed in the RWE studies. The results showed that the inclusion of RWE studies during the evidence synthesis process could increase the certainty of the estimates when the rare events meta-analysis of RCTs could provide enough evidence, but this depended on the method of inclusion and the confidence level placed in the RWE studies.

We do not recommend using NDS as the main method of analysis to combine rare events data. A recent scoping review found that the NDS approach was the most frequently used in empirical analyses [16]. However, NDS ignores differences in study designs and cannot consider the potential bias of RWE studies [20]. Furthermore, compared with RCTs, the results of RWE studies often show a large effect because of some uncontrolled confounding bias factors [53], and their interval estimates are much smaller because the events and the sample size are usually much larger [21]. Therefore, with the inclusion of RWE studies in a rare events meta-analysis of RCTs using NDS, not only can the bias of RWE studies not be adjusted but it would also give a larger weight than that of RCTs. Our two illustrative examples were also confirmed. This indicated that addressing the bias of RWE studies leads to more informed decision-making when integrating RCTs and RWE to assess the effects of rare events.

For the DAS, the data from the RWE studies are down-weighted based on the confidence levels of their credibility. An obvious feature of this model is that it needs to separate assessments of bias for each RWE study, which could be done by eliciting expert opinions regarding the bias parameters [54]. Another useful method is to gauge the impact on the model estimates by varying the amount of confidence placed in RWE studies [20]. For the RPI, there are two key differences with the DAS: one is in the estimation of heterogeneity that is performed separately for RCTs and RWE studies, and the other is that it only needs to set the magnitude of possible bias for the total RWE. Therefore, this model may be more attractive in empirical analyses, because one can not only set the prior distribution of heterogeneity parameters according to the characteristics of RCT and RWE studies (such as patient follow-up and statistical analysis methods), but can also set the magnitude of bias based on the risk of bias evaluation results for the whole RWE rather than for each RWE study.

Although the advantage of the THM is that it can obtain the estimates of each study design to increase its precision, the results of the total combined effect are conservative. In these two examples, compared to the results of the other models, the most notable difference was that the interval length was much larger when using the THM, even though we placed a high level of confidence in the RWE studies. This was because the THM explicitly considers the heterogeneity between study designs, allowing for additional variability across studies [20]. The main goal of the inclusion of RWE studies in a rare events meta-analysis of RCTs is to increase the power in testing whether the true effect exists. Rare events are often associated with safety results such as serious adverse events, indicating that the THM as the main method of analysis may not be suitable and may increase the chances of patient exposure to unnecessary danger [55]. The estimates can be improved by using expert opinions and meta-epidemiology to set a WIP for the combined effects [56]. For instance, constructing a WIP works via the consideration of the prior expected range of the treatment effect [57] and the heterogeneity values [58].

There were some limitations of this study that need to be recognized. First, this study only utilized two illustrative examples, the results may differ in other outcomes, and broader analyses in further empirical analyses are necessary. Second, this study only investigated methods for the inclusion of RWE in rare event meta-analyses of RCTs, and the performance of these methods was not explored. There is no clear rule for choosing the method to combine RCTs and RWE studies; when different methods lead to contradicting conclusions, researchers can choose those methods that lead to the desired outcome. Thus, a further evaluation of these methods in different scenarios, including the use of comprehensive simulation studies, is warranted. Third, meta-regression was not considered in this study. Although meta-regression can explain some between-study heterogeneity, it may be limited by the information on available covariates of each study or the number of studies in a meta-analysis. Fourth, although we used the OR as the effect measure, these methods are generalizable to other measures of association frequently used in meta-analyses, such as the RR, RD, and weighted mean differences. The performance of these methods in other measures needs to be further explored.

## 5. Conclusions

In summary, including RWE in a rare events meta-analysis has the potential to corroborate findings from RCTs, increase precision, and enhance the decision-making process, but this depends on the method of inclusion and the assumption for the magnitude of bias risk of RWE. The statistical performance of these approaches requires further evaluation in different scenarios of empirical or simulation studies.

## Figures and Tables

**Table 1 jcm-12-01690-t001:** Characteristics of included studies in the meta-analysis of the efficacy and effectiveness of pneumococcal polysaccharide vaccine (PPV23) vaccination against invasive pneumococcal disease (IPD) in elderly patients.

First Author (Year)	Design	Cases/Participants	Log (OR)	SE	Risk of Bias
Honkanen (1999) [25]	RCT	7/38,037	−0.97	0.84	Unclear
Maruyama (2010) [26]	RCT	3/2289	−1.94	1.51	Low
Ortqvist (1998) [27]	RCT	6/1666	−1.52	1.1	Low
Hechter (2012) [28]	Cohort study	9/31,282	−1.04	0.89	High
Jackson (2003) [29]	Cohort study	61/47,365	−0.58	0.26	Low
Ochoa-Gondar (2014) [30]	Cohort study	16/27,204	−0.96	0.75	Low
Tsai (2015) [31]	Cohort study	57/458,362	−1.42	0.33	High
Vila-Corcoles (2006) [32]	Cohort study	22/11,241	−0.51	0.51	Low
Dominguez (2005) [33]	Case-control study	149/596	−1.18	0.26	Low
Leventer-Roberts (2015) [34]	Case-control study	212/1060	−0.54	0.17	Low
Vila-Corcoles (2009) [35]	Case-control study	94/282	−1.08	0.34	Low

**Table 2 jcm-12-01690-t002:** Characteristics of studies included in the meta-analysis of the risk of diabetic ketoacidosis (DKA) among patients using sodium/glucose cotransporter 2 (SGLT-2) inhibitors compared with active comparators.

First Author (Year)	Design	Cases/Participants	Log (OR)	SE	Study Quality
Lavalle-González (2013) [38]	RCT	1/1284	−0.79	1.23	Good
Roden (2015) [39]	RCT	1/680	0.49	1.24	Good
Haering (2015) [40]	RCT	2/2702	−0.23	1.01	Good
Frías (2016) [41]	RCT	1/463	−0.70	1.23	Good
Hollander (2018) [42]	RCT	1/1361	0.50	1.24	Fair
Pratley (2018) [43]	RCT	1/1232	0.31	1.27	Good
Gallo (2019) [44]	RCT	1/414	0.70	1.23	Good
Fralick (2017) [45]	Cohort study	81/76,090	0.79	0.24	Good
Wang (2017) [36]	Cohort study	55/60,932	0.65	0.38	Fair
Kim (2018) [46]	Cohort study	63/112,650	−0.05	0.25	Good
Ueda (2018) [47]	Cohort study	30/34,426	0.76	0.38	Good
Wang-CCAE (2019) [48]	Cohort study	668/220,504	0.34	0.10	Good
Wang-MDCD (2019) [48]	Cohort study	155/20,532	0.17	0.20	Good
Wang-MDCR (2019) [48]	Cohort study	80/27,764	0.98	0.34	Good
Wang-Optum (2019) [48]	Cohort study	379/115,722	0.25	0.14	Good
Douros (2020) [10]	Cohort study	505/404,372	1.05	0.18	Good

**Table 3 jcm-12-01690-t003:** Estimated effects and 95% credibility intervals of the effectiveness of pneumococcal polysaccharide vaccine (PPV23) vaccination in preventing invasive pneumococcal disease (IPD) in elderly patients.

Variance Inflation Factor (*w*)	DAS	RPI	THM
*w*~beta (0.25, 1)	0.40 (0.22–0.67)	0.38 (0.12–0.91)	0.37 (0.09–1.32)
*w*~beta (1.5, 1)	0.42 (0.27–0.62)	0.40 (0.23–0.64)	0.39 (0.12–1.20)
*w*~beta (4, 1)	0.42 (0.28–0.59)	0.41 (0.26–0.61)	0.39 (0.14–1.14)

**Table 4 jcm-12-01690-t004:** Estimated effects and 95% credibility intervals of the risk of diabetic ketoacidosis (DKA) among users receiving sodium/glucose cotransporter 2 (SGLT-2) inhibitors versus active comparators.

Variance Inflation Factor (*w*)	DAS	RPI	THM
*w*~beta (0.25, 1)	1.46 (1.06–2.07)	1.41 (0.56–2.45)	1.32 (0.36–3.81)
*w*~beta (1.5, 1)	1.53 (1.19–2.00)	1.52 (0.98–2.19)	1.40 (0.47–3.73)
*w*~beta (4, 1)	1.56 (1.21–2.03)	1.56 (1.12–2.12)	1.42 (0.50–3.75)

## Data Availability

The computing code used in the study is provided in the Appendix A.

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
