# Peer review of "Methods for the Inclusion of Real-World Evidence in a Rare Events Meta-Analysis of Randomized Controlled Trials"

_jcm, 2023, doi:10.3390/jcm12041690_

Round 1
Reviewer 1 Report
Recently, there has been a growing interest in the use of real-world evidence (RWE) in drug clinical assessment and health care evaluation[4-6]. The term RWE needs to be more defined. Although they indicate references to explanatory studies, these are elusive and the reader may not know what the authors are referring to.
RWE studies may be subject to bias, due to lack of randomization[13]. Although the authors indicate the limitation of randomization, they should explain it so that the study can be understood without completing the reference.
the aim of this case study is to investigate whether these statistical methods, including the naïve data synthesis (NDS), the design-adjusted synthesis (DAS), the use of RWE as prior information (RPI), and the three-level hierarchical models (THM),could be used for integrating RWE in rare events meta-analysis of RCTs by considering the potential inherent biases of RWE studies and its impact on the level of uncertainty around the estimates.
In the last section they mix objective and design. The objective should be inside introduction, and design in material and methods
We understand that it is not a case study but a comparative case review study.
Author Response
Dear reviewer,
Thank you very much for giving us the opportunity to revise our manuscript. We are grateful to you for your insightful feedback and suggestions. The comments are very helpful for improving the manuscript. We have studied your comments carefully and have done our best to revise our manuscript according to the comments. Attached please find the response to the comments for your consideration.
We would like to express our great appreciation again for your comments and suggestions on our manuscript.
Best wishes to you.
Xin Sun
E-mail: sunxin@wchscu.cn
Responses to your comments:
Recently, there has been a growing interest in the use of real-world evidence (RWE) in drug clinical assessment and health care evaluation [4-6]. The term RWE needs to be more defined. Although they indicate references to explanatory studies, these are elusive and the reader may not know what the authors are referring to.
Response: We appreciate your suggestions. We have made some corrections, please see lines 64-66.
- RWE studies may be subject to bias, due to lack of randomization [13]. Although the authors indicate the limitation of randomization, they should explain it so that the study can be understood without completing the reference.
Response: We appreciate your suggestions. We made an explanation. Please see lines 78-79.
- the aim of this case study is to investigate whether these statistical methods, including the naïve data synthesis (NDS), the design-adjusted synthesis (DAS), the use of RWE as prior information (RPI), and the three-level hierarchical models (THM),could be used for integrating RWE in rare events meta-analysis of RCTs by considering the potential inherent biases of RWE studies and its impact on the level of uncertainty around the estimates.
Response: We appreciate your suggestions. We have made corrections, and removed “case”. Please see line 87.
Reviewer 2 Report
It is recommended to add the findings of all relevant studies such as "Methods for the inclusion of real-world evidence in network meta-analysis and applications of simple and accessible methods for meta-analysis involving rare events: A simulation study" and evidence gaps.
Author Response
Dear reviewer,
Thank you very much for giving us the opportunity to revise our manuscript. We are grateful to you for your insightful feedback and suggestions. We added the relevant studies based on your suggestions. Attached please find the response to the comments for your consideration.
We would like to express our great appreciation again for your comments and suggestions on our manuscript.
Best wishes for you.
Xin Sun
E-mail: sunxin@wchscu.cn
Reviewer 3 Report
In this manuscript, the authors stated that the real-world evidence inclusion has the ability to verify RCT findings, increase precision, and improve decision-making, however this is dependent on the technique of inclusion and the assumption about the level of bias risk of real-world evidence.
I have only 3 minor comments:
1. During the whole manuscript, the authors shold leave a space between the last letter and the reference number, according to the JCM instructions (no spaces for all), for example, the statistical power may be lower[3]....
2. There are inconsistent subheadings in the method section, mainly in the 2.3 section, consider using: 2.3.1......
3. Discussion section need to be proofread to ensure the flow and clarity for the readers
Author Response
Dear reviewer,
Thank you very much for giving us the opportunity to revise our manuscript. We are grateful to you for your insightful feedback and suggestions. The comments are very helpful for improving the manuscript. We have studied your comments carefully and have done our best to revise our manuscript according to the comments. Attached please find the response to the comments for your consideration.
We would like to express our great appreciation again for your comments and suggestions on our manuscript.
Best wishes to you.
Xin Sun
E-mail: sunxin@wchscu.cn
Responses to your comments:
During the whole manuscript, the authors should leave a space between the last letter and the reference number, according to the JCM instructions (no spaces for all), for example, the statistical power may be lower[3].
Response: We appreciate your suggestion. We have made corrections following your suggestion. And leave a space between the last letter and the reference number.
- There are inconsistent subheadings in the method section, mainly in the 2.3 section, consider using: 2.3.1......
Response: We appreciate your suggestions. We renamed subsection 2.3.
- Discussion section need to be proofread to ensure the flow and clarity for the readers.
Response: We appreciate your suggestions. We proofread the Discussion section based on your suggestions.
Reviewer 4 Report
The manuscript presents a description of a study comparing 4 methods of synthesizing effects from observational and mixed observational and clinical studies, on two examples of meta-analyses, with the aim of determining the method that provides the best estimate. The correction of study results depending on their design proved to be the safest and simplest for application in practice, which is of great importance to guide readers to the application of that particular method. The study is methodologically sound, with fairly stated limitations. The style in which the manuscript is written is clear, the English language requires only minor corrections.
Author Response
Dear reviewer,
We would like to express our great appreciation for your comments on our manuscript.
Best wishes to you.
Xin Sun
E-mail: sunxin@wchscu.cn
Round 2
Reviewer 2 Report
-